# Role of the Mediator Complex and MicroRNAs in Breast Cancer Etiology

**DOI:** 10.3390/genes13020234

**Published:** 2022-01-26

**Authors:** Edio Maldonado, Sebastian Morales-Pison, Fabiola Urbina, Lilian Jara, Aldo Solari

**Affiliations:** 1Programa de Biología Celular y Molecular, Instituto de Ciencias Biomédicas, Facultad de Medicina, Universidad de Chile, Santiago 8380453, Chile; emaldona@emed.uchile.cl (E.M.); fabi.urbina1516@gmail.com (F.U.); 2Programa de Genética Humana, Instituto de Ciencias Biomédicas, Facultad de Medicina, Universidad de Chile, Santiago 8380453, Chile; seba.morales.p@gmail.com (S.M.-P.); ljara@uchile.cl (L.J.)

**Keywords:** MED1 coactivator, MED12 coactivator, breast cancer, miRNA regulation

## Abstract

Transcriptional coactivators play a key role in RNA polymerase II transcription and gene regulation. One of the most important transcriptional coactivators is the Mediator (MED) complex, which is an evolutionary conserved large multiprotein complex. MED transduces the signal between DNA-bound transcriptional activators (gene-specific transcription factors) to the RNA polymerase II transcription machinery to activate transcription. It is known that MED plays an essential role in ER-mediated gene expression mainly through the MED1 subunit, since estrogen receptor (ER) can interact with MED1 by specific protein–protein interactions; therefore, MED1 plays a fundamental role in ER-positive breast cancer (BC) etiology. Additionally, other MED subunits also play a role in BC etiology. On the other hand, microRNAs (miRNAs) are a family of small non-coding RNAs, which can regulate gene expression at the post-transcriptional level by binding in a sequence-specific fashion at the 3′ UTR of the messenger RNA. The miRNAs are also important factors that influence oncogenic signaling in BC by acting as both tumor suppressors and oncogenes. Moreover, miRNAs are involved in endocrine therapy resistance of BC, specifically to tamoxifen, a drug that is used to target ER signaling. In metazoans, very little is known about the transcriptional regulation of miRNA by the MED complex and less about the transcriptional regulation of miRNAs involved in BC initiation and progression. Recently, it has been shown that MED1 is able to regulate the transcription of the ER-dependent miR-191/425 cluster promoting BC cell proliferation and migration. In this review, we will discuss the role of MED1 transcriptional coactivator in the etiology of BC and in endocrine therapy-resistance of BC and also the contribution of other MED subunits to BC development, progression and metastasis. Lastly, we identified miRNAs that potentially can regulate the expression of MED subunits.

## 1. Introduction

MicroRNAs (miRNAs) are small non-coding RNAs of 17–25 nucleotides in length that control gene expression in a post-transcriptional fashion. They are master regulators of gene expression, which control gene expression via either messenger RNA (mRNA) translational repression or mRNA degradation [1,2]. Those miRNAs are key players of regulatory biological mechanisms operating in several organisms, which includes development, host–pathogen interactions, cell differentiation, proliferation, apoptosis, metabolism, and tumorigenesis [1,2]. The estimated number of miRNAs can reach nearly 1–5% of all predicted genes in flies, nematodes, and mammals [1,2]. A large number of miRNA genes are scattered through the genome; however, some of them are found in clusters and co-expressed as polycistronic units displaying a functional relationship. More than half of the miRNAs are located within introns of their host genes, and they are co-expressed with their neighboring protein-coding genes [1,2]. Studies have shown that most mammalian miRNAs are organized in transcription units with their own RNA polymerase II promoter [3,4,5,6,7]. However, a genomic analysis of miRNAs in the human chromosome 19 miRNA cluster (C19MC) has revealed that those miRNAs are interspersed between Alu repeats and Alu transcription is dependent on the RNA polymerase III transcription machinery. Therefore, it was found that Alu elements upstream of the C19MC miRNAs retain promoter sequences involved in RNA polymerase III transcription. By chromatin immunoprecipitation and cell-free transcription assays it was demonstrated that RNA polymerase III is able to transcribe the C19MC miRNAs [8]. Therefore, the C19MC miRNAs are expected to constitute the 3′ end of Alu transcripts when expressed. This evidence indicates that some of the miRNAs can be transcribed by RNA polymerase III in primates [8]. The transcription units are known as miRNA clusters, and they contain multiple miRNAs, which are not separated from each other by a transcription unit [3,4,5,6,7]. MiRNAs transcribed by RNA polymerase II are initially synthesized as primary miRNA (pri-miRNA) transcript consisting of one or more hairpin structures, which are processed in the nucleus to a second structure called pre-miRNA. This pre-miRNA structure is then processed by the endoribonuclease DICER in the cytoplasm to a mature miRNA [1,2]. The pri-miRNA transcripts contain 5′cap structure and can undergo splicing and polyadenylation and generate more than one mature functional miRNA [1,2]. The miRNAs are subject to tight regulation at multiple steps, mainly transcription, processing, and localization. Additionally, feedback regulation loops exist between miRNAs and transcription and coactivators factors, which results in another layer of control mechanisms [1,4].

Dysregulation of miRNA clusters can contribute to the pathophysiological aspects of breast cancer (BC) and genetic and epigenetic changes are responsible for abnormal miRNA cluster expression. In this article, we will mainly review the role of Mediator complex subunits in BC and the role of MED1 in regulating the expression of an oncogenic miRNA cluster. We will also summarize the role of miRNAs in regulating the expression of MED complex subunits.

## 2. Role of miRNA in Breast Cancer

MiRNAs are known to be aberrantly expressed in cancer, including breast cancer (BC), and some of them can act as tumor suppressors and others functioning as oncogenes (oncogenic miRNA) which depends on the gene or pathway they regulate. Abnormal expression of miRNAs can affect the expression of tumor suppressors, transcription factors, transcriptional coactivators and oncogenic protein-coding genes and lead to the transformation of normal cells to tumor cells and subsequent metastasis [4,6]. They have been shown to be involved in every complex cellular process from cell cycle to apoptosis to cell migration and invasion, which indicates their importance in normal cellular function and disease. MiRNA dysregulation in BC can occur both at the genetic and epigenetic levels via the introduction of single-nucleotide polymorphism (SNP) into the miRNA sequence itself, or within the miRNA target binding sites, or via aberrant DNA methylation and histone modification [9,10]. Those SNPs within the miRNA target binding sites can reduce or to completely abrogate the ability of the miRNA to bind a target mRNA, therefore the regulation of that gene is lost. It is known that aberrant DNA methylation contributes to BC development and progression [11]. Promoter methylation of miRNA clusters can inhibit miRNA expression and also hypermethylation of CpG islands near the transcription start site repress the expression of miRNAs [11]. Besides methylation of DNA promoter sequences of miRNA, histone modifications can also regulate miRNA expression in BC. For example, histone methylation can downregulate miRNA expression in BC [12]. There exist several miRNAs that are dysregulated in BC through both genetic and epigenetic mechanisms, and the impact of the dysregulation on BC development and progression as well as the involvement in resistance to therapies can be found in the reviews by Kandettu et al. [5] and Mulrane et al. [11]. MiRNA cluster dysregulation also contributes to the pathophysiological aspects of BC and studies using several models have shown abnormal expression of miRNA clusters, which contribute to BC development and progression. The main dysregulated miRNA clusters in BC are summarized in Table 1 (Oncogenic miRNA clusters) and Table 2 (Tumor-suppressor miRNA clusters). A detailed description of those miRNA clusters can be found in the review by Kandettu et al. [5].

## 3. Molecular Classification of Breast Cancer

BC are heterogeneous and show variable morphological and biological features, therefore present different clinical behavior, and response to the treatment. Since the underlying genetic alterations and biological events, which are involved in cancer development and progression are quite complex, a cancer classification can provide an accurate diagnosis and a better prediction of tumor behavior to facilitate oncologic treatment. The classification includes molecular markers expression such as estrogen receptor alpha (ERα), progesterone receptor (PR) and human epidermal growth factor receptor 2 (HER2) [13]. The global gene expression profiling studies have classified BC in four intrinsic subtypes by hierarchical clustering, which are luminal A, luminal B, HER2-overexpressing, and basal-like BC (see Table 3) (reviewed in reference [13]). The gene profile and molecular features of the different subtypes of BC can be seen in Table 3. Luminal A ER+ HER2-BC represent 50–70% of invasive BC and they are low grade and have the best prognosis among all the intrinsic subtypes. Luminal B ER+ BC represent about 20–30% and tend to be higher grade and have worse prognosis than luminal A, and they show lower expression of ER-related genes, however, show higher expression of cell proliferation-related genes and a variable expression of HER2. The HER2 overexpressing BC subtype comprises about 15% of all invasive BC and shows high expression of HER2 and HER2 signaling-associated genes and also an overexpression of those genes located in the HER2 amplicon on chromosome 17q12 [13]. Those HER2 overexpressing BC are ER−, tend to be high grade and display an aggressive clinical course. They do not respond to endocrine therapy, but they are highly responsive to the anti-HER2-targeted therapy. Finally, those BC that belong to the basal-like BC show expression of genes expressed in normal mammary basal/myoepithelial cells, which includes cytokeratins [14]. Those BCs show overexpression of cell proliferation-related genes; however, it lacks ER, PR and HER2 and they are also named triple negative BC [13]. They are high grade with high cell proliferation index and the patients have poor prognosis and relapses can occur within 5 years after initial diagnosis [13,14].

Recent advances in high-throughput technologies have refined the classification of BC. The detailed molecular information of genomic alterations might aid the prognostication and risk stratification, thus leading to a better treatment of BC patients. Additionally, the molecular information might provide an opportunity for novel targeted treatments, which can be directed to the underlying molecular dysregulations driving individual BC and tumor growth. Detailed molecular characterizations of BC tumors can be found in a recent article by Abd-Elnabi et al. ([15] and references therein).

## 4. Role of Estrogen Receptor aplha in Breast Cancer Development

The ERα, also named estrogen receptor 1 (ESR1, referred as ER, since it also exits a related estrogen receptor beta), is a key functional mediator of estrogen signaling pathway and plays a prominent role in BC development [16,17,18,19,20,21]. Over 70% of BC are ER+ and estrogen is the primary driver of BC initiation, progression, and metastasis [16,17,18,19,20,21]. However, other drivers of BC initiation might also exist ([20] and references therein). The ER belongs to the nuclear steroid hormone receptors superfamily, which upon binding of the hormone in the cytoplasm are able to translocate to the nucleus and act as a transcription factor that bind to the gene promoters containing hormone responsive elements (HRE), which leads to the expression of those genes which can regulate different cellular and physiological processes [18]. The ER binds to specific DNA-binding regions of the gene promoter of target genes, which is called estrogen responsive element (ERE) and regulates gene expression [18,19]. However, the ER also regulates gene expression without directly binding to DNA [21]. This can occur through protein–protein interactions with other transcription factors into the nucleus [21]. In addition, membrane-associated ER initiates estrogen-dependent signaling pathways, which can lead both to altered functions of proteins in the cytoplasm and to regulation of gene expression [21]. The latter two mechanisms of ER action enable a broader range of genes to be regulated than the range that can be regulated by the classical mechanism of ER action alone. Most of the ER-target genes are involved in cell growth and cell proliferation and therefore any alteration causing ER activation will result in the expression of those genes leading to the activation of downstream signaling pathways [18,19]. Therefore, the ER is one of the main targets for endocrine therapy of ER+ BC (reviewed in [22]).

The ER is structured in several different domains [22] (Figure 1), which include the N-terminal domain (NTD; A/B) with a transactivation function, the DNA binding domain (DBD; C), which plays an important role in receptor dimerization necessary for ER function. The hinge region (H; D) consists of a flexible region and connects the DBD and the ligand binding domain (LBD; E). The LBD is the estrogen binding site, and is required for receptor dimerization, nuclear localization, and transcriptional coactivators/corepressors recruitment. The C-terminal domain (F) contributes to the transactivation capacity of the receptor. When the ER-estrogen complex binds to gene promoters through the ERE, it can recruit a variety of transcriptional coactivators such as the p160/SRC, p300/CBP and PCAF family of proteins, which function as histone acetyltransferases to modify chromatin status and facilitate the access of the transcription machinery (RNA polymerase II and the GTFs) to the gene promoters, which results in the gene activation of the target gene [23,24]. Depending on the conformational state of the bound receptor this can recruits co-repressors of transcription such as N-CoR and SMRT, which results in repression of the target gene [23,24,25] (Figure 2).

As mentioned earlier, the ER is critical in determining the human BC phenotype and is one of the main therapeutic targets. Activation of the ER is responsible for various biological processes, including cell growth, cell differentiation and programmed cell death [26,27,28]. Additionally, it has been reported that the response of ER to estrogens is critical in controlling transcription of specific ER-target genes [18,19,28]. Results from several studies have revealed that dysregulation of ER contributes to BC initiation, progression and also to therapeutic resistance and metastasis [18,19,29]. Nearly 70% of the diagnosed BC are ER+ and ER signaling alteration is one of the defining and driving events that contributes to tumor growth and BC progression in those tumors. The main dysregulation events of ER contributing to BC initiation and progression are: (i) increasing of the transcriptional activity of ER in an estrogen-independent fashion, (ii) gene amplification of the ER gene, (iii) point mutations in the ER gene, and (iv) genomic rearrangements of the ER gene (reviewed in reference [29]).

ER is a transcription factor and downstream signaling events from aberrantly activated growth factor tyrosine kinases such as epidermal growth factor receptor (EGFR) and HER2 can phosphorylate and increase ER transcriptional activity in a hormone-independent fashion [29]. Those ER+ BC tumors with amplified HER2 have a reduced ER expression, therefore have a reduced sensitivity to ER-targeted endocrine therapy and poor clinical outcomes [29]. Additionally, a dysregulation of cell cycle components is a common feature in ER+ BC, especially the Cyclin D-Cyclin Dependent Kinase (CDK)4/6-Retinoblastoma (Rb) axis in the luminal B subtype [29,30]. In those luminal B subtypes of BC, amplification of the Cyclin D gene occurs and gene copy gain of CDK4, together with a loss of negative regulators of CDKs, such as p16 and p18. Those events in conjunction with downstream activity from HER2 and EGFR can promote Rb phosphorylation and endocrine therapy resistance [29].

ER gene amplification is found in nearly 30% of the ER+ BC tumors, which depends on the detection method and scoring system [29]. Additionally, an important number of ER+ BC harbors ER gene copy number gain. Those events result in ER protein overexpression, which indicates that ER gene amplification present in early-stage BC can drive BC progression. Interestingly, studies have shown that a subset of ER+ BC with amplified ER gene are associated with tamoxifen resistance and poor prognosis [31,32,33,34]. Conversely, other studies have found that ER gene amplification is indicative of a longer disease-free survival and an increased sensitivity to tamoxifen endocrine therapy [29]. Those conflicting results might indicate that another driver gene could play an additional role in those different subsets of ER+ BC tumors. Therefore, more studies are required to better understand the functional significance of ER gene amplification in BC tumorigenesis and the role of this amplification in triggering endocrine therapy resistance and metastasis.

The acquisition of activating point mutations clustering within the LBD of the ER confers constitutive, estrogen-independent activity of ER in BC [35]. Those point mutations are Y537S or Y537N and D538G and arose from treatment of BC patients with aromatase inhibitors, since they were not found in matched primary samples [29]. Additionally, it has been found that in another group of patients with metastatic ER+ BC harbored mutations in the LBD, including Y537S, Y537C, Y537N, D538G and L336Q. All these patients were exposed to serial endocrine therapies and the matched primary samples did not harbor the mutation. Those results indicate that those acquired point mutations arose after the endocrine therapy resistance and metastasis. Several experimental models of ER gene point mutations have shown that ER LBD mutants can drive estrogen-independent cell proliferation, which is resistant to tamoxifen treatment [29].

Besides point mutations in the ER LBD, structural rearrangements in the ER gene have been reported and several ER gene fusion transcripts have been identified in luminal BC tumors [36,37]. For example, it has been reported that analysis from BC tissue samples, all of the luminal B subtype, almost 2% of the examined samples contained a recurrent fusion transcript that involved the first two non-coding exons of ER gene fused to various C-termini sequences from the coiled-coil domain containing 170 gene (CCDC170) [37]. Those gene fusions do not contain sufficient coding sequences to generate a chimeric ER fusion protein, but instead generate truncated forms of CCDC170. The exogenous expression of the truncated forms of CCDC170 in ER+ BC cells results in enhanced growth and tamoxifen resistance, which suggest a role for ER-CCDC170 fusions in endocrine therapy resistance [29,37]. Several other ER fusion transcripts have been reported and they are summarized in reference [29]. A study has reported a somatic gain-of-function event presented in the form of chromosomal translocation, which produced an in-frame fusion gene consisting of exons 1–6 of ER and the C-terminus of the Hippo pathway coactivator gene YAP1, which generates a stable ER fusion protein, resulting in a highly active constitutive transcription factor [29,38,39]. This translocation was identified in a patient which presented a highly aggressive endocrine therapy resistant and metastatic ER+ BC. In that fusion, the LBD was replaced with in-frame sequences from the other gene, therefore the estrogen-binding domain of ER is absent, and the drug used in the endocrine therapy cannot bind to ER. Those fusions promoted cell proliferation and constitutively activated transcription of ER target genes. Additionally, the fusions upregulated an epithelial-mesenchymal-transition-like transcriptional signature, induced cell motility and lung metastasis [29,40]. Interestingly, ER fusion-driven cell growth can be suppressed by CDK4/6 inhibition, suggesting that targeting kinases downstream of ER could be a strategy to treat patients with ER translocated BC tumors.

## 5. Mediator and Breast Cancer

Another transcriptional coactivator which plays a pivotal role on the ER-dependent transcription is the Mediator complex (MED) [41,42]. This complex was originally identified in budding yeast [43,44] and soon after in mammalian cells [45,46]. Mammalian MED is a large multiprotein complex that plays a key regulatory role in the RNA polymerase II transcriptional process. As mentioned, it was first discovered in yeast as a factor required for activator-dependent transcription of genes but is clear that is involved in every step of transcription by RNA polymerase II including preinitiation complex formation, transcription activation, initiation, promoter clearance, elongation, splicing, gene looping and termination [47,48]. Metazoan MED is composed of at least 30 different subunits named MEDs (from 1–30) and it is highly conserved among all eukaryotes [47]. The different transcriptional activators (also named specific transcription factors) can make specific contacts with one or more MED subunits, which are essential to finally transduce different cell signals and lead to gene activation [41,49]. It has been shown that Mediator is involved in different processes such as cell differentiation, organ development and mutations or alterations in the expression of its subunits can cause several diseases, including cancer [41,49,50]. Mediator functionally bridges promoter-bound transcription factors to the basal transcription machinery, thereby recruiting and activating RNA polymerase II transcription [41,49].

MED subunits are arranged in four different modules, known as Head, Middle, Tail and Kinase modules [41,42] (Figure 3). The Head, Middle and Tail modules form the core MED, while the Kinase module associates to it when required. It is considered that the Kinase module is involved in transcriptional repression, since it can phosphorylate the C-terminal domain of the largest subunit of RNA polymerase II (CTD) and some other components of the transcriptional machinery [41,49]. Since it is a quite large complex, its subunits provide interacting surfaces for protein–protein interactions with transcriptional activators and proteins involved in the transcription process. Some specific subunits can interact with DNA-bound transcriptional activators, therefore transducing the signal from the DNA-bound activators to the transcription machinery acting as a bridge between the two of them.

Nowadays, transcriptional machinery malfunction has been shown that elicits broad effects on cell proliferation, development, differentiation, and pathologic disease states, including cancer. Indeed, the aberrant activation of specific genes is a result of a malfunction of the transcription machinery. In this section, we will focus on the correlation of MED subunits and BC, especially on MED1, MED7, MED12, MED15, MED19, MED23, MED24, MED27 and MED28, which have been shown to play a role in BC initiation, progression, and metastases [50].

### 5.1. MED1–MED24

The first known link between MED and cancer was the association of MED1 and BC in breast cancer tissues and cell lines [51]. MED1 expression positively correlated with the epidermal growth factor receptor 2 (HER2) status of the tumors, and it is highly phosphorylated in a HER2-dependent manner [50,51]. MED1 is located in the Middle module of MED complex (Figure 3) and plays a key role in BC initiation and progression, since interacts with the ER to activate the expression of ER-target genes [41,50,51,52,53]. As for the mechanism of expression of ER-target genes, the ER interacts mainly with the MED1 subunit of the mammalian Mediator complex through directly binding to the LxxLL motifs (also named NR-boxes; L correspond to Leucine and x correspond to any amino acid residue) of this polypeptide in a ligand-dependent manner, thereby recruiting and activating the RNA polymerase II transcription machinery. It is a polypeptide of 1.581 amino acids long with several functional domains as described in Figure 4A. MED1 mRNA and MED1 protein have been found to be overexpressed in about 60% of all primary BC and BC cell lines [51]. Additionally, MED1 gene in humans locates in chromosome 17 into the 17q12 region, which is known as the “HER2 amplicon” [54,55]. MED1 co-amplifies with HER2 in essentially all BC examined and the expression level of MED1 correlates with HER2 status [51]. On the other hand, it has been demonstrated that the knockdown of MED1 gene abolishes the expression of ER-dependent reporter genes and the endogenous ER-dependent genes, but not the expression of the genes that are controlled by other transcriptional activators, such as p53 [52,53]. Moreover, knockdown of MED1 gene impairs not only the expression of these ER-target genes, but also the estrogen-dependent growth of BC cells [52,53]. Additionally, it was found that a MED1 LxxLL mutant knock-in mouse model (the three L were changed to A in MED1 protein) is not affected in the fertility and survival of the mice, indicating that MED1 LxxLL motifs are not essential for mice survival [56]. However, it was found that MED1 LxxLL motifs were required in pubertal mammary gland development and play an important role in the mammary luminal progenitor cell formation and differentiation [56].

To examine whether MED1 participates in HER2-mediated breast tumorigenesis in vivo, Yang et al. [57] crossed the MED1 LxxLL-mutant knock-in mice with mammary tumor-prone MMTV-HER2 mice whose tumorigenesis was driven by the oncogene HER2. The results showed that MED1 LxxLL motif mutations can delay mammary tumor onset by an average of about 16 weeks when compared to normal controls. Moreover, the tumors of MED1-mutant mice grew slower and also had an overall lower tumor weight when compared to the controls [57]. Tissue staining of the lung tumors of the MED1-mutant mice had a loss in the number of metastatic lung nodules. The isolated MED1 LxxLL mutant tumor cells were not able to migrate, and they had not invasion capabilities. Furthermore, those mutant cells had reduced the number of cancer stem cell formation [57]. Further analysis revealed that those mutant MED1 LxxLL mutant tumor cells had a lower expression of both traditional ER-target genes including IGF-1 and cyclin D1, and HER2 activated ER-target genes such as LIF and ACP6 [57]. Recently, Yang et al. generated MED1 mammary-specific overexpression mice and crossed them with MMTV-HER2 mice [58]. It was found that MED1 overexpression can promote onset, growth, metastasis, and multiplicity of HER2-aggressive tumors in mice [58]. Moreover, the studies revealed that MED1 overexpression can promote epithelial-to-mesenchymal transition, cancer stem cell formation, and resistance to anti-HER2 therapy of the tumors derived from MED1 overexpressing mice [58]. A protein Jab1, a component of the COP9 (Constitutive photomorphogenesis 9) signalosome, has been identified as a key direct target of MED1 [58]. Jab1 can reciprocally regulate the stability and transcriptional activity of MED1 by controlling its ubiquitin-proteasome pathway-mediated turnover and its cyclic recruitment to target gene promoters [58]. All findings from those studies highlighted the role of MED1 and its LxxLL motifs as a key determinant in HER2-mediated BC tumorigenesis. This study also indicates that MED1 and its LxxLL motifs are promising tissue-specific therapeutic targets since the interference of the function of these motifs will inhibit gene expression of both ER-target genes and HER2 activated ER-target genes. The interference of MED1 function will stop or delay cell proliferation and thus BC tumor progression and metastases.

MED24 is a MED subunit that locates in the Tail submodule (Figure 3), and studies have shown that MED24 functionally communicates with MED1 to regulate pubertal mammary gland development, since mammary glands from MED1/MED24 double knockout mice showed a profound retardation in ductal branching during puberty [59]. Cells derived from both basal and luminal from knockout mice had impaired DNA synthesis and the expression of ER-targeted cyclin D1 and E2F1 was inhibited. In this study, it was also found that several BC cell lines have high levels of MED1 and MED24 and suppression of the expression of those proteins by siRNA inhibits DNA synthesis and cell growth [59]. Those results indicate that MED1 and MED24 functionally communicate to mediate ER function and cell growth in normal mammary gland cells and BC cells.

ARGLU1 (arginine and glutamate rich 1) colocalizes with MED1 in the nucleus, and directly interacts with a far C-terminal region of MED1. Studies indicate that ARGLU1 is able to cooperate with MED1 to regulate ER-mediated gene transcription. ARGLU1 is recruited, in a ligand-dependent manner, to ER-target gene promoters and is required for their expression [60]. By ChIP-reChIP assay, it was demonstrated that ARGLU1 and MED1 colocalize on the same ER-target gene promoter upon estrogen induction. Depletion of ARGLU1 by shRNA knockdown, significantly impairs the growth, and anchorage-dependent and -independent colony formation of BC cells [60]. Those results indicate that ARGLU1 as a new MED1-interacting protein required for estrogen-dependent gene transcription and BC cell growth.

### 5.2. MED7

This MED subunit is located in the Middle part of the MED complex (Figure 3) and has an important role in gonadal development and embryogenesis [61]. The role of MED7 has been studied in ER+ BC and it was found that high MED7 mRNA and protein expression is associated with good prognostic factors and improves BC-specific survival in patients with ER+/luminal subtypes [62]. The role of MED7 within the ER-controlled pathways might be very complex and it could be dependent on the MED7-specific interacting partners. In the study [62], MED7 expression was negatively associated with EGFR expression. When the EGRF is overexpressed in BC, it can increase tumor size and can worsen patient outcomes [63]. Additionally, EGFR overexpression negatively correlates with ER status [63]. Activation of EGFR by epidermal growth factor (EGF) can trigger a kinase cascade which can phosphorylate S118 of the NTD domain of ER, resulting in ER transactivation [63]. It is possible that MED7 may reduce the EGFR-mediated ligand-independent ER activation that occurs in BC.

### 5.3. MED12

MED12 is an extensively studied subunit of the mediator complex kinase module. Its link with numerous illnesses has attracted much attention [64,65,66]. MED12 changes in its sequence and expression may be harmful to cells, ultimately manifesting into different disease characteristics [67]. The MED12 gene is found on chromosome X, at location Xq13.1 encompassing a 25-kb area with 45 exons and coding for 230-kDa protein 2.177 amino acids long [64,65,66] (Figure 4B). It is part of the kinase module that consists of MED12, MED13, Cyclin C (CCNC) and cyclin-dependent kinase 8 (CDK8) [68] (Figure 3). CCNC-CDK8 can interact with MED12, which in turn interacts with MED13 to be recruited into the mediator complex [66,68] (Figure 3). The MED12 gene is expressed in all tissues [65]. The MED12 protein has numerous motives and domains as protein interaction sites regulating the transmission of diverse signals and those are described in Figure 4B. MED12 features is a proline-glutamine-leucine-rich C terminus, also known as the PQL domain, where many signal transduction pathways culminate, resulting in transcription regulation. For example, the β-catenin transactivation domain binds directly to MED12 PQL domain to induce target gene expression. MED12 C terminus ends in an odd-paired (OPA) motif domain. Both of these domains are crucial for the regulatory function of MED12, since they serve as binding sites for several transcription factors [48]. The protein also has a LCEWAV motif towards its N terminus, whose function remains unclear yet [65] (Figure 4B). Another remarkable aspect of MED12 N terminal is the presence of two overlapping LxxLL motifs, a characteristic binding site for nuclear hormone receptors [65] (Figure 4B). A MED12 paralog is also seen in cells, which is hypothesized that occurred via gene duplication. The paralog known as MED12-like is localized to chromosome 3q25.1. Note that MED12 and MED12L proteins are found mutually exclusively in the CDK8 kinase module. MED12 is a critical element of numerous cells signaling pathways and governs a wide spectrum of cell activities from cell lineage determination to carcinogenesis [64,65,66].

In BC, CDK8 increases estrogen-target gene expression in ER+ BC tumors [69]. Small molecule inhibitors of CDK8 (senexin B) have been proven to suppress cell cycle and tumor growth in ER+ BC models in vivo xenograft [69,70,71]. Mutations in MED12 impact CCNC-CDK8 kinase activation and disrupt the roles of mediator complex kinase assembly. However, BC MED12 mutations have also been demonstrated to deregulate the ER pathway. JMJD6 is a JmjC domain containing protein that binds to ERα-responsive enhancers and stimulates estrogen-responsive gene expression during estrogen stimulation [72]. Studies of chromatin immunoprecipitation indicated that MED12 interacts directly with JMJD6 to control estrogen signals in BC cells [65,72]. Knockdown of either JMJD6 or MED12 significantly suppresses estrogen-responsive genes [72]. Another study indicated that CARM1 (coactivator associated arginine methyltransferase 1) produces MED12 methylation on preserved arginine residues (R1862 and R1912), resulting in MED12 chromatin binding [73]. The MED12 methylation reduces the production of p21 via epigenetic regulation in BC cells, rendering them more vulnerable to chemotherapy [73]. Quantitative mass spectrometry investigation demonstrated that CARM1 needs JMJD6 for MED12 methylation, and its knockdown lowers the interaction between MED12 and CARM1 [72]. Overall, MED12 was revealed to be crucial for signaling estrogen in BC, promoting tumorigenesis.

### 5.4. MED15

This is a component of the MED Tail module (Figure 3), and it is known that plays key roles for signaling pathways. MED15 is overexpressed in several cancer types, including BC among others, and correlates with the clinical outcome and the recurrence of the disease [50,74]. However, the molecular mechanism by which MED15 overexpression contributes to those malign diseases remains unknown yet, although it is assumed that they are the result from an increased and sustained transcriptional activation. In HaCaT cells, an immortalized human keratinocyte cell line, the MED15 knockdown can attenuate the tumor growth factor B (TGFB)-induced gene expression and relieves the TGFB-mediated growth inhibition [74]. It has been proposed that MED15 regulates the TGFB/Smad signaling pathway [74]. On the other hand, MED15 knockdown can decrease the metastatic potential of a highly aggressive BC cell line by inhibiting the TGFB/Smad signaling pathway [74]. Taken altogether, those results indicate that one of the molecular mechanisms by which MED15 overexpression works is through the activation of the TGFB/Smad signaling pathway.

### 5.5. MED19

MED19 is a component of the Middle module of the MED complex (Figure 3). This polypeptide is associated with the development and progression of several cancer types, including BC as well [50]. Previous studies have shown that MED19 is upregulated in human BC tissues and the knockdown by RNA interference significatively suppresses the growth of BC cells [75,76]. The detailed mechanism by which MED19 can promote BC progression and the molecular mechanism of MED19 dysregulation in BC are unknown yet. However, it is possible that MED19 is a crucial transcriptional regulator of genes involved in cell cycle, cell proliferation, and epithelial-mesenchymal transition. Recently, it was found that MED19 levels are elevated in BC tissues and associated with larger tumors, a high grade of malignant features and poor prognosis [77]. Moreover, MED19 overexpression can enhance BC cell proliferation, cell invasion, epithelial-mesenchymal transition, and cell migration both in vivo and in vitro [77]. Additionally, MED19 can interact with EGFR and activates the EGFR/mitogen-activated protein kinase (MEK)/extracellular signal-regulated kinase (ERK) signaling pathway, which in turn induces carcinogenesis and BC progression [78].

### 5.6. MED23

This subunit has been shown to be part of the Tail module of MED complex (Figure 3) and it is a critical coactivator for the expression of ER-dependent target genes and growth of the estrogen-dependent BC cells [78]. MED23 is a binding target of ER and can bind to ER through two LxxLL motifs present at the N-terminus of MED23 and bridges the RNA polymerase II transcription machinery to regulate the expression of ER-dependent target genes [78]. It also participates in the formation of tamoxifen-resistance, a drug used to treat ER+ BC and its high expression is associated with poor prognosis of BC patients [78]. Recently, it has been shown that silencing of MED23 significantly inhibits cellular growth and proliferation of BC cell lines (BT474 and MCF-7) and renders everolimus-resistant BC cells sensitive to the treatment with this drug [78]. The silencing of MED23 in combination with everolimus treatment inhibits cell cycle progression of BC cells and can inhibit cell invasion and metastasis of BC [78]. Everolimus is a drug that is mainly used to treat refractory and metastatic BC and it has strong affinity for ER. This drug can effectively inhibit ER-driven gene expression, degrade the ER protein, and downregulate ER levels [79]. Unfortunately, drug resistance can occur in many patients during the treatment [79]. Based on those observations, MED23 can be used as a target for molecular therapy, since the silencing of MED23 inhibits BC tumorigenesis and overcomes drug-resistance.

### 5.7. MED27

This polypeptide is part of the Tail module of the MED complex. It is located in the junction of the Head and Tail of the MED complex (Figure 3) and interacts with MED29, which in turn interacts with MED14 to link the Tail module with the Head and Middle modules (Figure 3). Though the specific biological functions of MED27 are unknown yet, it clearly plays an essential role in early embryonic and neuronal development [80,81]. MED27 has been found that is highly expressed in BC tissues and cells and its expression correlates with tumor size and grade and the high expression of MED27 had a poor prognosis [82]. Downregulation of MED27 by transfection of MDA-MB-231 BC cells with si-MED27 results in a reduction in the levels of MED27 and the transcription factor Sp1 and also cell proliferation was suppressed, while cell apoptosis was enhanced [82]. Those results indicate that MED27 affects Sp1 expression and is probably that MED27 functions as a transcriptional cofactor for Sp1 activation. Sp1 exists in abundance in the nucleus of a variety of cells and tissues and its abnormal expression and activation can enhance tumor growth and metastasis.

### 5.8. MED28

MED28 is part of the Head module of the MED complex and plays important roles in transcriptional activation, since it interacts with multiple signaling molecules including the Grb2, Src family proteins and actin cytoskeleton [83]. In addition, MED28 has been found to be highly expressed in several cancer types, including prostate, colorectal and BC as well [50,84]. MED28 overexpression can stimulate cell proliferation and its suppression inhibits the tumor growth of BC cells and in a mouse xenograft model [85]. In addition, MED28 regulates both cell growth and cell migration in human BC cells [86]. The inhibition of MED28 expression can inhibit cell migration and is coincident with lower expression levels of MMP2 and MEKI. Conversely, the overexpression of MED28 augments MEKI-mediated MMP2 expression and cell migration in BC cells (MCF-7 cell line) [86]. On the other hand, the ectopic expression of cDNA encoding MMP2, or MEKI can rescue the inhibitory effect of MED28 or MEKI knockdown on cell invasion of the BC cells [86]. Recently, it was shown that in MDA-MB-231 BC cell line, the suppression of MED28 expression attenuates the mesenchymal morphology and downregulates the NFkB transcription factors, together with a reduction in mesenchymal biomarkers [83]. In the MCF-7 BC cell line, the administration of Adriamycin, an epithelial-mesenchymal transition-inductive system, can reduce cell–cell contacts and the cells show a fibroblast-like appearance [83]. Those morphological changes were correlated with the expression of MED28 and epithelial and mesenchymal markers and also the augment of the NFkB transcriptional activity [83]. Those observations indicate that Adriamycin can acts through the MED28/NFkB axis on Snail, which is a transcriptional repressor of E-cadherin, and its downstream mesenchymal biomarkers [83]. MED28 modulates the development of epithelial-mesenchymal transition though the NFkB/Snail axis in BC cells, therefore plays an important role in BC progression, cell growth and migration of BC cells.

## 6. Role of MED1 in the Resistance of Breast Cancer to the Endocrine Therapy

HER2 amplification and overexpression have been recognized as major contributors to the endocrine therapy resistance of BC [87,88,89], however, the underlying molecular mechanisms of action are not fully understood yet. HER2 receptor belongs to the EGF family of transmembrane tyrosine kinase receptors [90] and its own expression is dependent on both ER and MED1 [91]. It is also known that HER2 overexpression in BC cells is one of the principal mechanisms that contribute to endocrine therapy resistance to tamoxifen, a drug which is used to treat ER-positive BC [87,88,89,90,92,93].

Previous studies have reported MED1 to be phosphorylated and activated by the MAP kinase pathway at two key threonine residues (T1032 and T1457, Figure 4A) [94]. It is known that MAP kinase pathway is a key downstream pathway in the HER2 signaling cascade and it has been found that phosphor-MED1 levels are significantly higher in HER2-positive BT474 BC cells than in MCF-7 BC cells and the overexpression of HER2 in MCF-7 cells is sufficient to increase MED1 phosphorylation [95]. Most importantly, it was found that both HER2 and MAP kinase inhibitors were able to disrupt this HER2-mediated phosphorylation of MED1, indicating that MED1 is phosphorylated by MAP kinases (most likely ERK1 and ERK2, Figure 4A) [95]. On the other hand, MED1 knockdown re-sensitized these HER2-positive BC resistant cells to tamoxifen treatment [95]. At the molecular level, it was observed that in HER2-overexpressing cell lines, the co-activator MED1 is heavily phosphorylated, and it is preferentially recruited to the gene promoter of ER-responsive gene (TFF1) in the presence of tamoxifen, instead of the transcriptional co-repressors N-CoR and SMRT, an event leading to transcriptional activation of the ER-responsive genes [95] (Figure 2). Conversely, in those BC cell lines tamoxifen-sensitive, the co-repressors N-CoR and SMRT are preferentially recruited to the TFF1 gene promoter, which in turn results in inhibition of gene expression of ER-responsive genes (Figure 2). Moreover, the mutation of the MED1 phosphorylation sites (T > A) on T1032 and T1457 in those BC cell lines overexpressing HER2 and tamoxifen-resistant, results in restored tamoxifen-sensitive and tamoxifen-induced N-CoR and SMRT recruitment [95]. Additionally, in another study it has been reported that MED1 knockdown rendered the otherwise resistant BC cells sensitive to another anti-estrogen drug, fulvestrant, both in vitro and in orthotopic xenograft mouse models [96]. From all that data, we can conclude that MED1 plays a key role as a point of crosstalk between HER2-signaling and ER-signaling pathways and plays a role in HER2-mediated resistance to the endocrine therapy of BC. The data from all those studies are summarized in Figure 2. According with this idea, the overexpression of MED1 has been associated with poor treatment outcome, and the high MED1 expression correlates with poor survival of BC patients that have been treated with endocrine therapy and did not respond to this therapy [97,98]. Studies by Nagpal et al. [98] have demonstrated that MED1 gene is amplified in 10% of the patients with invasive BC and it is mutated in 0.6% of the patients with invasive BC. On the other hand, MED1 is upregulated in almost 20% of BC patients. The expression of the MED1 mRNA is significantly higher in ER+ and ER− BC patients compared to the expression in normal breast tissues [98]. When the correlation between MED1 mRNA levels and BC prognosis was examined in patients using the TCGA (The Cancer Genome Atlas https://portal.gdc.cancer.gov/ (5 January 2022)) database, it was found that BC patients with high levels of MED1 mRNA had a poorer survival than patients with low levels of MED1 mRNA and it is more significant for those patients with ER+ BC [98]. Those results indicate that MED1 is a key player in BC and its expression correlates with poor survival in those patients. Another intriguing study has revealed an increased frequency of MED1 mutations in the circulating tumor DNA in BC patients following endocrine therapy and anti-HER2 treatments [99]. Altogether, the data from these studies demonstrate that MED1 could be used as target for ER-positive BC treatment and to overcome the endocrine therapy resistance (prevalent in BC) and could improve the treatment outcomes of those BC patients.

## 7. MED1 Regulation of ER-Dependent Oncogenic miRNA in Breast Cancer

MED1 is involved in human breast carcinogenesis and treatment response and thus understanding its functioning is imperative. This coactivator is overexpressed in BC and is a negative prognostic factor. It has been shown that the oncogenic miRNA cluster, miR-191/425, is estrogen/ER regulated miRNAs, which are able to promote cell proliferation, cell migration and chemoresistance in ER-positive BC [98]. On the other hand, MED1 has been demonstrated to be a regulator of several miRNAs known to be involved in BC such as miR-10b-5p, -100-5p, -17-5p, 18a-5p, -191-5p, 193b-3p, 205-5p, -326, -422a and -425 suggesting its importance in BC pathogenesis. MED1 is able to induce the miR-191/425 cluster in an ER-dependent manner [98]. Occupancy of MED1 on EREs upstream of miR-191/425 cluster is estrogen and ER-dependent and ER-induced expression of these miRNAs is dependent on MED1 coactivator in MCF-7 BC cells [98]. An increase in the expression of genes involved in cellular proliferation and migration (JUN, FOS, EGFR, VEGF, MMP1, and ERBB4) is observed in response to MED1 overexpression along with miR-191 overexpression, however, a treatment with anti-miR-191 and MED1 overexpression is able to inhibit the expression of those genes involved in cell proliferation and migration [98]. Therefore, the MED1-mediated cellular functions are in part mediated through miR-191 and this miRNA is a downstream effector of MED1 function in BC. Additionally, MED1 also regulates the levels of direct miR-191 target genes such as SATB1, CDK6 and BDNF [98]. SATB1 is a transcription factor and a chromatin remodeler key in BC pathogenesis. It has been demonstrated that targeted downregulation by miR-191 is required for MCF7 cells to enhance cell proliferation and migration [98]. BDNF is a potent neurotrophic factor that stimulates BC cell growth and metastasis via tyrosine kinase receptors TrkA, TrkB, and the p75NTR death receptor. Overall, the results show that MED1/ER/miR-191 axis can promote BC cell proliferation and migration and it might be useful as a novel target for therapy. Moreover, MED1 was also shown to have significant positive correlation with the levels of miR-191-5p, miR-425-5p, miR-422a and miR-100-5p in BC patients [98]. The underlying molecular mechanisms by which miR-191 can mediate breast carcinogenesis have not been completely elucidated yet. A model of regulation for the Med1/ER/miR-191/425 axis is presented in Figure 5.

## 8. MiRNA Regulation of MED Subunits

MED1 has been demonstrated to be an important regulator of murine placental development [100]. When changes in miRNA expression were analyzed in human placental trophoblasts exposed to hypoxia, it was found that miR-205 is up-regulated and it can target MED1, since it interacts with a specific target at the 3′-UTR sequences of MED1 mRNA and silences its expression. This finding indicates that MED1 and miR-205 play a role in trophoblast injury [100]. In prostate cancer cells miR-205 transcription is repressed, due to hypermethylation of the MIR-205 locus, which leads to a decrease in miR-205 expression and therefore to an increase in MED1 expression, since as mentioned earlier, miR-205 is able to target MED1 mRNA and reduces its expression [101]. The overexpression of miR-205 in prostate cancer cells affects cell viability in a negative fashion, which suggests that miR-205 has a tumor suppressor function [101]. Interestingly, miR-205 was found to be downregulated by MED1 in BC cells, indicating that a negative feedback loop exists, which has not been studied yet [98].

Additionally, MED1 is a target for miR-146a, and in hepatic cells the overexpression of this miRNA improves glucose and insulin tolerance as well as the lipid accumulation in the liver by promoting the oxidative metabolism of fatty acids. miR-146a overexpression also increases the number of mitochondria and promotes mitochondrial respiration in hepatocytes [102]. Consequently, inhibition of miR-146a expression can significantly reduce the mitochondrial number and the expression of mitochondrial respiratory genes, however, the restoring of MED1 expression can abolish the effects of miR-146a on lipid metabolism and mitochondrial function [102]. Those results indicate that part of the miR-146a functions on lipid metabolism and mitochondrial function are through MED1.

Obesity is associated with increased cancer risk, predominantly cancers of digestive organs and cancer of hormone sensitive organs in women, including hormone receptor-positive/HER2-negative BC, which shows a relation between worst outcome and obesity [103]. The underlying molecular mechanisms that obesity/hyperleptinemia might cause a reduction in the efficacy of endocrine treatment have not been determined yet. Leptin can induce nuclear translocation of phosphorylated ER increasing the expression of ER-responsive genes and reduces tamoxifen-mediated gene repression by inhibiting tamoxifen-induced recruitment of transcriptional co-repressors such as NCoR and SMRT, while potentiates co-activator binding, including MED1 [104]. Recently, it has been shown that leptin can upregulate MED1 expression by targeting and decreasing miR-205, a miRNA that targets MED1 [104]. Additionally, leptin is able to increase its functional activation via phosphorylation, which is carried out by HER2 and EGFR. MED1 silencing abrogates the negative effect of leptin on tamoxifen efficacy, while honokiol and adiponectin (both antioxidants) treatments can inhibit leptin induced MED1 expression, improving tamoxifen efficacy [104]. Those results indicate that there are a leptin-miR-205-MED1 and leptin-HER2-EGFR-MED1 axes, which can contribute to the understanding of the molecular mechanisms of tamoxifen resistance in obesity/hyperleptinemia states.

Target predictions based on the miRDB (MicroRNA Target Prediction Database) database indicates that human oncogenic hsa-miR-96-5p is one of the most likely candidates to target MED1 at the 3′-UTR and indeed, it has been shown that MED1 is a direct target of the miRNA cluster miR-96/182/183 in the liver of diabetic rats treated with colesevelam [105]. MED19 is targeted by miR-101-3p and miR-422a in BC, since its expression negatively correlates with the expression of those miRNAs in BC tissues, and directly targets the 3′-UTR of MED19 mRNA [77]. In osteosarcoma cells, MED27 is a direct target of miR-18a, since it can bind to MED27 mRNA, and the expression of this miRNA suppresses tumor growth in mice [106].

As for MED7, MED12, MED15, MED23, MED24 and MED28, we were unable to find a miRNA which can directly target those coactivators, however, it seems likely that miRNA regulation on those MED subunits might exist, since they are key elements in the regulation of transcription, cell differentiation, development and carcinogenesis. A list with the miRNAs predicted to target the MED subunits is presented below:hsa-miR-3198-MED7;hsa-miR-5692a-MED12;hsa-miR-6165-MED15;hsa-miR-4282-MED23;hsa-miR-8485-MED24;hsa-miR-3613-3p-MED28.

## 9. MiRNA-Based Therapies for Breast Cancer

The growing evidence indicates that miRNAs are involved in BC initiation, progression, and metastasis, therefore it might be possible to either suppress or restore the expression of the BC-associated miRNA. In those cases, where reduced miRNA expression drives BC, this miRNA can be delivered to the cells to restore its function and expression and in circumstances where the miRNA is upregulated, an anti-miRNA can be delivered to counterattack its activity. However, a safe delivery of miRNAs or anti-miRNAs to target specific cells or tissues is still a major challenge for miRNA-based therapies. The main limitations which are associated with miRNA delivery are the nuclease attack of the RNA, rapid blood clearance, low tissue permeability and immunotoxicity [107,108]. Advances have been made to improve the stability and protection of those molecules against nuclease attack by using chemical modifications on the chains [109]. Additionally, methods have been developed to improve tissue penetration and enhance stability of the molecules [108]. As for miRNA-based therapies for BC, mainly two delivery systems have been used to deliver miRNA, which are the lipid-based and exosome-based delivery systems.

Lipid-based nanocarriers have been widely used to deliver nucleic acids into the cells. Cationic lipids contain hydrophilic heads and hydrophobic tails, which can easily form a complex with anionic nucleic acids resulting in a lipoplex that has high affinity for cell membranes and is not immunogenic. Those nanocarriers have been used to deliver [110,111]. The main limitation of the lipoplex is the low efficiency to deliver miRNA in vivo, however, the conjugation of a polyethylene glycol group to the lipoplex improves the overall efficiency of miRNA delivery in vivo [112,113]. For example, De Antonellis et al. [114] delivered miRNA 199b-5p into several BC cell lines through a stable lipoplex and they were able to inhibit cell proliferation by down regulation of Hes-1 expression.

Cell-secreted extracellular vesicles (EVs) are micro-sized and nano-sized membrane vesicles derived from several cell types and play key roles in cell–cell communication. Over the last two decades have become a very studied issue, since EVs are critical mediators in cell–cell communication in normal and pathological biological processes [115,116]. EVs can be classified in three major groups, which are micro vesicles (MVs), exosomes and apoptotic bodies [115,116]. MVs have a diameter of 100–1000 nm and are released by outward budding and fission of the plasma membrane. On the other hand, exosomes are smaller than MVs and have a diameter ranging from 30–150 nm and they are originated from intracellular multivesicular bodies and released into the extracellular environment by a fusion of the multivesicular bodies with the cell membrane [115,116]. MVs contain diverse bioactive molecules, such as nucleic acids, miRNAs or mRNAs, lipids, and proteins from the original cell type in which they originated [117,118]. Exosomes also contain bioactive molecules from the parental cells, similarly as the contents of the MVs [117,118]. Since EVs contain diverse bioactive molecules, which can be trafficked in between cells, they can be used as delivery platforms for therapeutic uses in the treatment of several diseases, including BC treatment [119,120]. Ohno et al. were able to deliver let-7a miRNA to EGFR-expressing xenograft BC tissue in mice by using exosomes and it was shown that the treatment is able to inhibit tumor development in vivo [121]. Those results indicate that exosomes can be therapeutically used to target BC tissues. Direct encapsulation of cargos into exosomes often relies on physical procedures including incubation, freeze–thaw cycles, electroporation, sonication, or membrane permeabilization. Exosomes can be engineered and loaded with different bioagents to treat several diseases, for example Huntington’s disease [122], brain inflammatory disease [123] and many cancer types [119,124,125]. Indeed, a clinical trial using engineered exosomes to target metastatic pancreatic cancer has been recently registered (NCT03608631). Exosomes can be loaded with miRNAs by calcium chloride-mediated electroporation [126] or by direct electroporation [124]. Direct electroporation has been used to incorporate anti-tumor miRNAs (miR-31 and miR-451a) to silence anti-apoptotic genes and therefore promote apoptosis in HepG2 hepatocarcinoma cell line [124]. Moreover, a recent study using Taxol-loaded exosomes from mesenchymal stroma/stem-like cells and applied to BC cells demonstrated that those Taxol-loaded exosomes can produce growth inhibition on the tumor cells by a cytotoxic mechanism [127]. Taxol-loaded exosomes can also inhibit tumor growth in vivo in mice and reduce metastases in other organs as well. Therefore, it is tempting to speculate that exosomes loaded with a combination of a BC tumor suppressor miRNA and Taxol could effectively inhibit tumor growth and metastases in BC. A systematic review on miRNA-based therapeutics in BC, which summarizes the current knowledge on miRNA-based treatments for BC and mainly focused on in vivo models can be found in reference [128].

## 10. Perspectives

BC is the most common cancer in women worldwide and despite the advances in treatment still is the leading cause of death by cancer among women. Drug resistance, BC subtype heterogeneity and tumor relapse have hampered the effectiveness of BC therapy. Therefore, new therapeutic targets and new therapies against BC must be developed and in this regard miRNA-based therapies offer an excellent alternative. However, the use of miRNA-based therapies against BC still is an undeveloped field. Additionally, therapies with small molecules aimed to interfere with the ER-MED1 or another ER-MED subunit interaction might be valuable tools to treat BC.

In this regard, therapies to inhibit the activity of the miR-191/495 cluster would be able to inhibit BC development and progression or alternatively a therapy that overexpress miR-205 would have benefits to reduce the amount of MED1 coactivator, which is a key player in BC development and progression. Nowadays, the direct delivery of RNA molecules into the cell cytoplasm is possible, since mRNA-based vaccines have been developed to fight the SARS-CoV-2 coronavirus, which causes the COVID-19 disease. Another useful therapy would be the search or development of small molecules to interfere with the interaction of the ER with the LxxLL motif, which is crucial in transcriptional activation by the ER. Those small molecules must be cell permeable and be able to travel to the nucleus to exert their function. Thus, in the next decade we will be able to obtain new therapeutic treatments against BC, which will save lives and reduce the deaths caused by BC.

Lastly, caution must be taken since we must be aware that BC is a very complex and heterogeneous disease and miRNA-based therapies seem to be attractive since those molecules can simultaneously modulate multiple dysregulated genes and/or cellular pathways. Therefore, miRNA molecules possess a pleiotropic role and care should be taken until all targets of each miRNA will be known in order to be used as therapeutic molecules and they can be completely manageable for safe clinical purposes.

## Figures and Tables

**Figure 1 genes-13-00234-f001:**
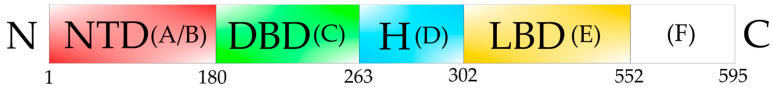
Schematic representation of the human Estrogen Receptor alpha. The NTD domain (N-terminal domain, red, (**A**/**B**)) contains the sub-domains (**A**,**B**), which are essential for transcriptional activation. DBD domain (DNA-binding domain, green, (**C**)) is the specific DNA-binding domain, which binds to the ERE in the gene promoter of ER-target genes. The H region is the flexible hinge domain (blue, (**D**)). LBD domain (ligand binding domain, yellow, (**E**)) is the specific estrogen binding site, and is required for receptor dimerization, nuclear localization, and transcriptional coactivators/corepressors recruitment. (**F**) domain (white) seems to perform a similar function as the NTD domain. The numbers indicate the amino acid residue positions in the ER.

**Figure 2 genes-13-00234-f002:**
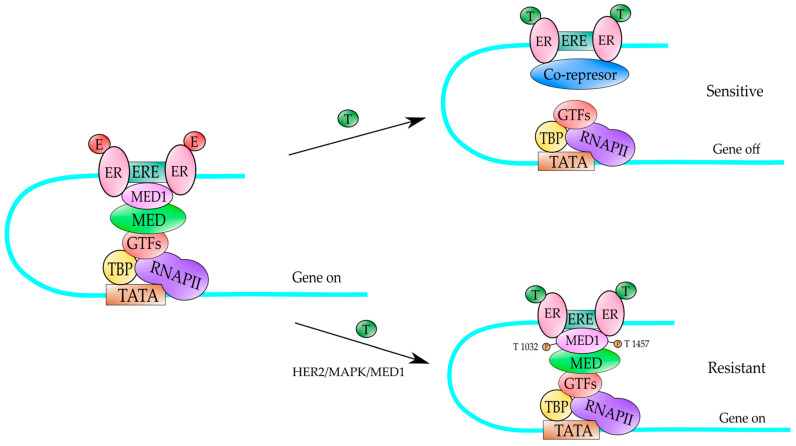
A mechanism to explain the tamoxifen-resistance in ER-positive HER2-expressing BC. Under normal cellular conditions, the estrogens (E) can bind to the ER and activate gene expression of ER-target genes (gene on). In those ER-positive BC, the tamoxifen (T) binds to the ER and produces a conformational change that results in the recruitment of transcriptional co-repressors (SMRT and N-CoR) which block gene expression (gene off) of ER-target genes, and it produces a tamoxifen-sensitive BC. However, in the presence of HER2 receptor, the ER receptor bound to tamoxifen (T), recruits MED1 co-activator, which is heavily phosphorylated by MAPK kinases (activated by HER2) producing an activation of those ER-target genes (gene on) and the BC becomes tamoxifen-resistant. MAPK kinases phosphorylate MED1 at T residues in position 1032 and 1457. TATA is the DNA-binding sequence of TBP factor in the gene promoter. RNAPII is the RNA polymerase II enzyme which transcribes. MED is the Mediator complex. GTFs are the auxiliary factors for RNAPII and ERE is the estrogen response element.

**Figure 3 genes-13-00234-f003:**
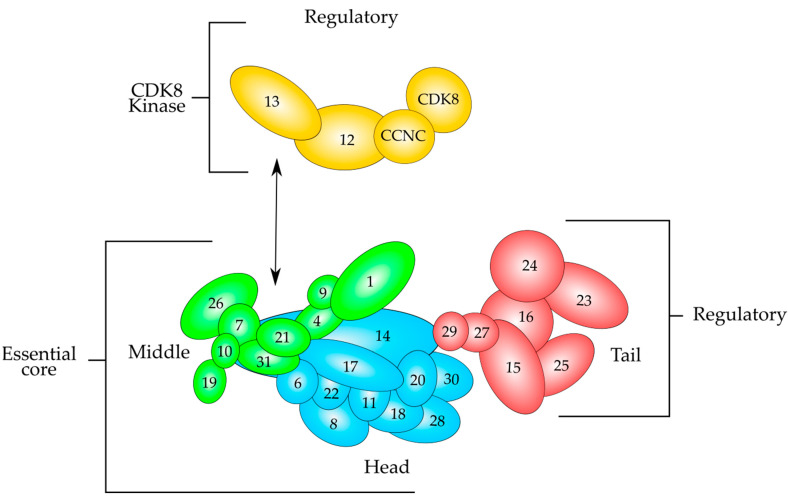
Schematic overview of the modular structure of the mammalian Mediator complex based on recent evidence derived from electron microscopy data. The Mediator consists of almost thirty polypeptides organized in three modules, which are Tail (red), Middle (green) and Head (cyan), which are held together by MED14. The Head and Middle modules constitute the essential core that is necessary for transcription regulation, whereas the Tail and the CDK8 kinase modules perform regulatory functions. The kinase module (yellow) consists of MED12, MED13 and the CCNC-CDK8 pair. This module associates to the Mediator after cell signaling and is able to repress transcription. CCNC-CDK8 pair can interact with MED12, and this polypeptide interacts with MED13, which tethers the Kinase module to the Mediator by interacting mainly with MED26 and MED1 of the Middle module. Modified from reference [41].

**Figure 4 genes-13-00234-f004:**
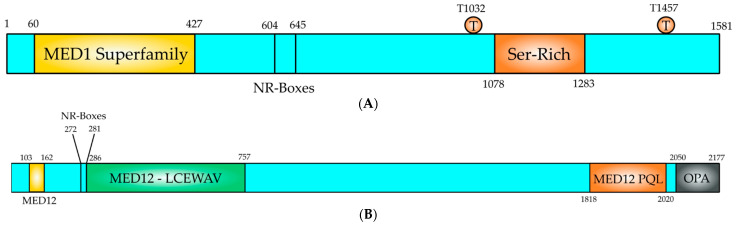
Structural domain organization of MED1 and MED12 proteins. (**A**) MED1 is a 1581 amino acids long polypeptide, which contains functional domains to perform its coactivator function. The MED1 domain (yellow, 60–427) is conserved in MED1 proteins from all eukaryotes. The NR-boxes or LxxLL motifs locate at positions 604 (LTSLL) and 645 (LMNLL) and they serve as protein–protein interaction motifs for nuclear receptors, such as ER. A Ser-Rich domain is located at the C-terminus of the polypeptide (orange, 1078–1283). Two residues (T1032) and (T1457) are phosphorylated by MAPK, an event that increases the activity of MED1, (**B**) The N-terminal region contains the CCNC-CDK8 binding and activation domain (1–50) and contains also a MED1 conserved region (yellow, 103–162). Two overlapping NR-boxes (272–281; LLKLLPLL), which are nuclear receptor binding LxxLL motifs. A MED12-LCEWAV domain (green, 286–757) is present in the polypeptide, which is unknown yet. At the C-terminus contains a Proline-Glutamine-Leucine (orange, 1818–2020; MED12-PQL), which binds effector molecules. At the C-terminal end of the polypeptide is located the odd-paired domain (black, 2250–2177; OPA), which also binds effector molecules.

**Figure 5 genes-13-00234-f005:**
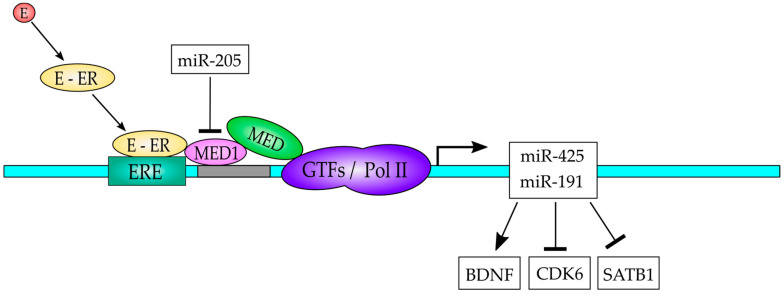
A model to explain the interplay of the ER-MED1-miR-191/425 in BC. The estrogen (E) binds to the estrogen receptor (ER) and translocate to the nucleus, where it can bind to the estrogen receptor element (ERE) in the gene promoter region of the miR-191/425 gene to activate its expression. MED1 is recruited to the promoter by interactions with the ER and serves as a bridge between promoter-bound ER and the general transcription factors (GTFs)/pol II transcription machinery. The expressed miR-191/425 regulates downstream target genes as indicated.

**Table 1 genes-13-00234-t001:** Oncogenic MicroRNA Cluster, their Chromosomal location and members of MicroRNA Cluster.

MicroRNA Cluster	Chromosomal Location	MicroRNAs in the Cluster	Model Systems
miR-221/222	Xp11.3	miR-221, miR-222	Human tissue sample and cell lines (MCF7, MDA-MB-231, MDA-MB-453, and SKBR3)
miR-23a/27a/24-2	19p13.12	miR-23a, miR-27a, miR-24-2	Human tissue sample
miR-23b/27b/24-1	9q22.23	miR-23b, miR-27b, miR-24-1	Human tissue sample
miR-106b/25	7q22.1	miR-106b, miR-93 and miR-25	Human tissue sample and cell line (MCF-7)
miR-106a/363	Xp26.2	miR-106a, miR-18b, miR-20b, miR-19b-2, miR-92-2, miR-363	Human tissue sample and cell line (JIMT-1 and KPL-4)
miR-200c/141	12p13.31	miR-200c, miR-141	Cell lines (MCF-7, BT474 and T47D)
miR-301b/130b	22q11.21	miR-301b, miR-130b	Human tissue sample
miR-532/502	Xp11.23	miR-532-5p, miR-188-3p, miR-362-3p, miR-362-5p, miR-501-3p, miR-660-3p, miR-502-3p, miR-502-5p	Human tissue sample
miR-191/425	3p21.31	miR-191-5p, miR-191-3p, miR-425-5p, miR-425-3p	Cell line (MDA-MB-231-luc)
miR-371/373	19q13.4	miR-371, miR-372, miR-373	Human tissue samples

**Table 2 genes-13-00234-t002:** Tumor suppressor MicroRNA Cluster, their Chromosomal location and members of MicroRNA Cluster.

MicroRNA Cluster	Chromosomal Location	MicroRNAs in the Cluster	Model Systezms
miR-199a/214	1q24	miR-199a-5p, miR-199a-3p and miR-214	Cell line (T47D and MDA-MB-231)
miR-212/132	17p13.3	miR-132, miR-212	Human tissue samples
miR-143/145	5q33	miR-145, miR-143	Cell line (MCF-7, SK-BR-3, and MDA-MB-231) and human tissue samples
miR-497/195	17p13.1	miR-195, miR-497	Cell lines (BOY and T24)
miR-200b/200a/429	1p36.33	miR-200b, miR-200a, miR-429	Cell line (MDA-MB-231 LM2)
miR-302/367	4q25	miR-367, 302d, 302c-5p, 302c-3p, 302a-5p, 302-3p, 302b-5p, 302b-3p	Cell line (MDA-MB-231 and SK-BR-3)
miR-15a/16	13q14.2	miR-15a, miR-16-1	Cell line (MCF-7 and MDA-MB-231)

**Table 3 genes-13-00234-t003:** Overview of different BC molecular subtypes.

Intrinsic Subtype	Gene Profile	Molecular Findings	IHC Phenotype	Histologic Subtypes	Integrative Cluster	DNA Architecture	Survival
Luminal A	High expression of luminal epithelial genes and ER-related genes	Mutations PI3KCA, MAPK3K1, and GATA3; CCDN1 amplification; no corresponding activation of PI3K pathway	ER+, PR ≥ 20%, HER−, Ki67low	Tubular Carcinoma, low-grade IDC-NST, classic ILC	IntClust 2	11q13/14 amplification; firestorm pattern of high-level copy number gains	Poor
					IntClust 3	Low genomic instability	Good
					IntClust 4	CNA devoid	Good
					IntClust 6	High genomic instability 8p12 amplification	Intermediate
					IntClust 7	16p gain, 16q loss, 8q amplification	Good
					IntClust 8	1q gain, 16q loss	Good
Luminal B	Lower expression of luminal eplithelium and ER-related genes, but higher level of proliferation and HER2− related genes that luminal A	Similar to luminal A but with a higher prevalence of TP53 and RB pathways inactivation as well as Myc-related and FOXM1 related transcription	ER+, PR < 20%/or HER2+/or Ki67high	IDC-NST, micropapillary carcinoma, pleomorphic ILC	IntClust 1	High genomic instability; 17q23 amplification; GATA3 mutation	Intermediate
					IntClust 2	See above	
					IntClust 5	HER2 amplification	Poor
					IntClust 6	See above	
					IntClust 9	8q gain, 20q amplification	Intermediate
HER2-OE	High expression of HER2-related genes; low expression of ER-related genes	HER2 amplicon and EGFR/HER2 signal protein signature	ER−, PR−, HER2+	High-grade IDC-NST, pleomorphic ILC	IntClust 5	See above	
Basal like	High expression of basal epithelial and proliferation genes; low expression of HER2-related and ER-related genes	Mutations in TP53; losses in RB1 and BRCA1; amplification of MYC; high PI3K/AKT pathway activation	ER−, PR−, HER−	High-grade UDC-NST metaplastic carcinoma, medullary carcinoma, adenoid cystic carcinoma	IntClust 10	5q loss, 8q gain, 10p gain, 12p gain; high genomic alterations with sawtooth pattern	Poor
					IntClust 4	See above	

ER indicates estrogen receptor; HER2, human epidermal growth factor receptor 2; IDC-NST, infiltrating duct carcinomas, no special type; IHC, immunohistochemically; ILC, invasive lobular carcinomas; PR, progesterone receptor; Ki67 antigen measures the proliferation status; Integrative Cluster is a breast cancer classification of 10 different subgroups with distinctive molecular profiles and clinical outcomes.

## Data Availability

All data are included within the manuscript.

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
