# Peer review of "Role of the Mediator Complex and MicroRNAs in Breast Cancer Etiology"

_genes, 2022, doi:10.3390/genes13020234_

Round 1

Reviewer 1 Report

This review aims to summarize the regulation of microRNAs by mediator in breast cancer, with a novel aspect yet a lack of comprehensiveness. In addition, the structure and flow need improvements.  

Major:

  1. This paper mainly focuses on the ER positive breast cancer. Although ER positive is about 70% of all breast cancers, HER2-Positive and Triple-Negative are more aggressive. Therefore, these subtypes should be reviewed as well.
  2. In the section 2, “Role of Estrogen Receptor in breast cancer development (Line69-122)”, the role of ER in breast cancer development was not summarized. Instead, the role of Mediator on ER expression was described, which should go to the section 3 “Mediator and BC”. This is a structural issue. Sections should be re-organized and modified to make the content match the subtitle.
  3. Page 5, line 164-165. “MED1 co-amplifies with HER2 in essentially all BC examined and the expression level of MED1 correlates with HER2 status”. Any human survival curve data available to show the correlation of MED1 expression and survival rate?
  4. In the section 3, several works on the role of MED1 in BC are missing. DOI:10.1074/jbc.M110.206029. DOI: 10.1158/0008-5472.CAN-17-1533. DOI: 10.1016/j.celrep.2021.108822. DOI: 10.1128/MCB.05245-11.
  5. In the section 3 “Mediator and BC”, only Med1 is introduced. Other subunits of Mediator should also be summarized. Detailed function of Med1 should be moved to the section 4, “role of MED1 in the resistance of BC to the endocrine therapy”.
  6. There are studies on the function of Med7, Med15, Med19, Med23, Med24, Med27, Med28 in breast cancer. The authors should include these works as well and put them in section 3.
  7. The structural domain organization of proteins can be combined to make one figure.
  8. The role of miRNA in breast cancer should be summarized as a section.

Minor:

Summary of the field or expression is not accurate. Should be checked carefully.

  1. Page2, Line 45-46. “However, findings have shown that most mammalian miRNAs are organized in transcription units with their own RNA polymerase II promoter.” Several miRNAs are transcribed by Pol III but not Pol II. Please add the information and citations.
  2. Page3 Line 101. “chromatin structure” is not accurate. HAT functions as modifier of histone. “chromatin status” or “chromatin accessibility”

There are many grammar and spelling mistakes in the manuscript. English language editing is needed. Here listed several examples:

  1. Page2, Line 58. “tumor-suppressive”, grammatical mistake. Correct expression: “tumor suppressor”.
  2. Page2 Line 67. “two subunits of the Mediator of transcription”. Better use “two subunits of the Mediator” or “two subunits of the Mediator that functions in transcription”.
  3. Page2 Line 73. “metastases”, grammatical mistake. Correct expression “metastasis”
  4. Page3 Line99-100. “When the ER-estrogen complex binds to gene promoters through the ERE can recruit a variety of transcriptional coactivators such as the p160/SRC, p300/CBP 100 and PCAF family of proteins”. Should be “When the ER-estrogen complex binds to gene promoters through the ERE, it can recruit a variety of transcriptional coactivators such as the p160/SRC, p300/CBP 100 and PCAF family of proteins”

Author Response

Reviewer 1, thank you for your suggestions: 

  1. We added a complete section (section 3) describing the different molecular subtypes of breast cancer, including a table (Table III) that describes the main characteristics of those subtypes.
  2. We summarized the role of estrogen receptor in BC in section 4, especially the role of estrogen receptor in therapy resistance and metastasis of breast cancer. We moved all information related to MED1 functions towards section 5.
  3. We included available data on MED1 overexpression and survival rate (Figure 5).
  4. We properly referenced those works, and they are highlighted in the reference section
  5. We added a figure (Figure 3) displaying the complete structure of the Mediator complex, obtained from cryo-electron microscopy to illustrate the individual subunits of this complex. Detailed MED1 functions on endocrine therapy resistance was moved to section 6.
  6. The functions of all those Mediator subunits in breast cancer were described in section 5.
  7. We combined MED1 and MED12 subunit structures in one figure (Figure 4), however, because ER structure is described in another section, we were unable to combine all of them in one figure.
  8. The role of miRNA in BC was summarized in section 2.
  9. 10. Those sentences were changed according to the reviewer suggestions

11.12.13. Spelling was corrected and also the text was extensively revised for any grammatical mistakes.

14.The sentence was changed according to the reviewer suggestion

Reviewer 2 Report

In the manuscript “Regulation of Oncogenic microRNAs Expression Involved in 2 Breast Cancer by Transcriptional Coactivators of the Mediator 3 Complex” Maldonado et al. present the role of  Mediator subunit 1 and 12 transcriptional coactivators involved in the development and progression of breast cancer. The potential role of miRNAs in the regulation of MED1 expression, as well as the possible employed of these miRNA in therapy, are also discussed, focusing on their possible delivery mechanism.

The topic is very interesting and current. The manuscript is well written. The figures were well prepared.

However, I should advise the Authors to introduce a small paragraph on the main deregulated miRNAs in breast cancer.

Author Response

Reviewer 2, thank you for your suggestions: 

  1. We included two tables (II and III) listing the main miRNA clusters that can act as oncogenes and tumor-suppressors in breast cancer.

Round 2

Reviewer 1 Report

The authors have sufficiently improved their paper, provided a detailed and thorough response to the comments and addressed my major concerns.

I have one additional comment for the authors to consider: In figure2, add phosphorylation of MED1 by MAPK kinases in the tamoxifen-resistance panel.

Author Response

  1. In figure 2, we added the phosphate groups on MED1 at the Thr residues in which is phosphorylated by MAPK.

  1. We have modified extensively figure 3, as long as we could, since we cannot change any positions or size of the individual MED subunits, as they have been located by cryo-electron microscopy studies.

  1. We have deleted figure 5, as we could not find a way to modify it, since it is a figure made from data mined from a database. Instead, we described in the text the correlation between MED1 overexpression and poor survival in breast cancer patients. We hope that would be clear enough for the readers.